# Safely Learning Optimal Auctions:
# A Testable Learning Framework for Mechanism Design

**Vikram Kher** [1]   **Manolis Zampetakis** [1]

## Abstract

When can the distributional assumptions of theorems and learning algorithms be trusted? Inspired by this question, Rubinfeld & Vasilyan (2023) initiated the study of *testable learning*. In this schema we always learn one of the following two things: either we have achieved the desired accuracy regardless of whether the distributional assumptions are satisfied, or the input distribution does not satisfy the original distributional assumptions. Motivated by the challenge of relying on strong distributional assumptions in many theorems in mechanism design, we develop a *testable learning framework for mechanism design*. Traditional models in mechanism design assume that value distributions satisfy some notion of *regularity*. Unfortunately, testing regularity is not possible in the original testable learning framework as we show. To bypass this impossibility, we propose a regularized version of the testable learning framework. Under this framework, we always learn one of the following two things: either we achieve high revenue compared to the best possible revenue of any regular distribution close to the input distribution, or the input distribution does not satisfy regularity. We then use this framework to provide: 1) a tester-learner pair for revenue-optimal mechanisms, 2) a tester for whether the fundamental Bulow-Klemperer Theorem (Bulow & Klemperer, 1996) is applicable to a given dataset, and 3) a tester to confirm the existence of an anonymous reserve price that results in the anonymous price auction securing a constant fraction of the optimal revenue.

---

[1]Department of Computer Science, Yale University, Connecticut, USA. Correspondence to: Vikram Kher <vikram.kher@yale.edu>.

*Proceedings of the 42nd International Conference on Machine Learning*, Vancouver, Canada. PMLR 267, 2025. Copyright 2025 by the author(s).

## 1. Introduction

Data science is a field with celebrated applications across many scientific disciplines. Many important statements and algorithms in data science require distributional assumptions on the data that is being analyzed. In physical sciences, a common distributional assumption is the Gaussianity of the underlying data, whereas in social sciences and economics more heavy-tailed distributions arise (Clementi & Gallegati, 2005). An instantiation of this phenomenon appears in learning theory as well, where strong computational impossibility results for agnostic learning prevent general learning guarantees and to bypass these lower bounds it is common to assume that the marginals are uniform or Gaussian (Kalai et al., 2008; Diakonikolas et al., 2021). A natural question to ask in all these settings is the following: *How can we infer which statements or algorithms can be applied in a given dataset?* A straightforward way to tackle this question would be to design a testing algorithm that can help us distinguish which of the distributional assumptions are satisfied by the given dataset. Unfortunately, directly verifying these assumptions is often computationally intractable (Goldreich, 2017). This brings us back to a dilemma for a given dataset: should we use methods that have stronger guarantees but require stronger distributional assumptions? Or, should we use methods that have weaker guarantees but are applicable in settings with fewer assumptions?

Motivated by this dilemma, the recent breakthrough work of Rubinfeld & Vasilyan (2023) introduced the framework of *testable learning* wherein a learning algorithm is paired with a *tester* which satisfies the following properties: (1) if the tester accepts, then we can be confident that the learning algorithm is applicable to our dataset, whereas (2) if the tester rejects, then we know that the distributional assumptions are not satisfied. This allows us to bypass the lower bounds of testing distributional assumptions because in the case that the tester accepts we cannot make any conclusions about the distributional assumptions; we only know that the learning algorithm succeeds. Rubinfeld & Vasilyan (2023) develops a computationally efficient tester-learner pair for agnostically learning half-spaces over Gaussians and the uniform hypercube. Many works have since followed that successfully apply this testable learning framework to a va-

riety of different settings (Diakonikolas et al., 2023; Klivans et al., 2024; Gollakota et al., 2023a;b; 2024).

In the context of mechanism design, there are distributional assumptions that underline the vast majority of results in single-item revenue maximization. When the distribution is explicitly known, Myerson (1981) effectively solved this question through his characterization of virtual valuation functions. For each bidder $i \in [n]$, who draws their value from a distribution with cdf $F_i(v)$ and pdf $f_i(v)$, Myerson (1981) defined their virtual value function to be

$$\phi_i(v) = v - \frac{1 - F_i(v)}{f_i(v)}.$$

If $\phi_i(v)$ is non-decreasing, then the underlying distribution is called *regular*. The use of regularity in mechanism design dates back to the celebrated work of Myerson (1981), and has remained a highly influential concept in auction theory ever since. Given that regularity is crucial to many results and methods in mechanism design, it is natural ask the following question:

> **Question:** *When is it **safe** to apply methods that are based on regularity in an auction setup?*

**Our Contribution.** In this work, our goal is to answer the question above by adapting the aforementioned framework of testable learning in the field of mechanism design. We first observe that a vanilla application of the testable learning framework of Rubinfeld & Vasilyan (2023) is not appropriate for mechanism design (see Section 3.1). Then, we introduce a relaxation of this framework that we call *Regularized Testable Learning*, which relaxes the learning guarantee that we have when the tester algorithm accepts. This new framework allows us to build a powerful tester that distinguishes whether many theorems and methods in revenue maximization are applicable to a given data source. We believe our framework will have applications beyond just the study of revenue-maximizing mechanisms.

### 1.1. Our Results

Our primary conceptual contribution is our *Regularized Testable Learning* framework. In regularized testable learning, we are informally guided by the following principles. Let $\mathcal{L}$ be a learner for the concept class $\mathcal{C}$ with respect to a family of distributions $\mathcal{D}$ and a hypothesis class $\mathcal{H}$. If $\mathcal{T}$ is a tester for the distributional assumptions of $\mathcal{L}$ then the following should hold:

▷ Soundness: If $\mathcal{T}$ accepts a distribution $D$, then w.h.p. $\mathcal{L}$ should output a hypothesis which achieves good objective performance on $D$ **relative to the optimum concept of any close distribution to $D$ in $\mathcal{D}$.**

▷ Completeness: If a distribution $D$ is close to some $D' \in \mathcal{D}$, then w.h.p. $\mathcal{T}$ should accept $D$ on input.

This is a relaxation of the testable learning framework of Rubinfeld & Vasilyan (2023), due to the highlighted bold part in the soundness condition. The original definition of soundness from Rubinfeld & Vasilyan (2023) requires the learner to achieve good objective performance *relative to the optimum concept of the distribution itself.* In our setting, the family of distributions $\mathcal{D}$ is regular distributions and the optimal concept of any distribution $D \in \mathcal{D}$ is its Myerson optimal mechanism (Definition 6). Furthermore, in our setting, two distributions are close to each other if they have a small Kolmogorov-Smirnov distance (Definition 9).

We formally introduce the regularized testable learning framework in Section 3. In Section 3 we also show a proof that the original testable learning framework cannot give us nontrivial results in the revenue maximization setting. We instantiate our framework in the following settings:

**(1) Learning Revenue-Optimal Mechanisms (Theorem 6).** Under this model, first proposed by Cole & Roughgarden (2014), the goal is to characterize how many samples are needed to achieve a $(1 - \epsilon)$ multiplicative approximation to the revenue of the optimal mechanism. In a unifying result, Guo et al. (2019) effectively settled the sample complexity of single item revenue maximizing auctions by providing tight upper and lower bounds (up to poly-log factors). Drawing an interesting connection with the work of Guo et al. (2021), we introduce a parameter $\alpha$ such that on an input distribution $D$ our learner should output a mechanism $M$ such that $Rev(M, D) \geq (1 - g(\alpha))OPT(D')$ for every regular distribution $D'$ that is $\alpha$-close in *Kolmogorov-Smirnov (KS) distance* (if the tester accepts). Importantly, if there is no $\alpha$-close regular distribution, then our learner bears no guarantee. We present these results in Section 5.

**(2) Competition vs Optimality (Theorem 8).** Bulow-Klemperer Theorem (Bulow & Klemperer, 1996) is a well-known result in auction theory that states that when the value distributions of the bidders are i.i.d. and regular, then the revenue of a simple Vickrey auction with one additional player is higher than the revenue of an optimal mechanism for one less bidder, that is, competition in the i.i.d. setting is more important than optimality. The proof of this result heavily relies on the fact that the value distributions are regular. In Section 6 we show that there is a regularity tester, based on samples from the value distributions, that: (1) when it accepts, then the conclusion of Bulow-Klemperer's Theorem holds, and (2) when it rejects, then the value distributions are certainly irregular. This gives a nice per-instance test of whether Bulow-Klemperer's result is applicable or not.

**(3) Anonymous Price Auctions (Theorem 10).** A well-

known result by Hartline & Roughgarden (2009); Alaei et al. (2019) states that when the value distributions are regular but not symmetric, the auction that sets an anonymous reserve price, i.e., a reserve price that is the same for all bidders, achieves a constant fraction of the optimal revenue of Myerson's auction. This result also relies heavily on the regularity of the value distributions, and hence it makes sense to ask if there is a way to test if this result is applicable or not. Similar to the previous case of the Bulow-Klemperer Theorem, we provide a tester that: (1) when it accepts, then anonymous price auctions achieve a constant approximation of the optimal revenue, whereas (2) when it rejects, then the value distributions are certainly irregular. We provide details on this in Section 7.

Finally, in Section 8 we show a single-bidder lower bound for the sample complexity of regularized testable learning for revenue maximization.

**Novelty of Results.** Under our framework, we obtain the first nontrivial revenue guarantees for learning single-item auctions when the bidders draw their values from *unbounded, irregular* distributions. In contrast, previous work on learning irregular auctions (e.g., Guo et al. (2019) and Roughgarden & Schrijvers (2016)) delivers sample complexity results that scale with an upper bound on the support of the distribution. By dropping this dependence, our framework is applicable to many new settings, such as when the underlying distribution is a mixture of Gaussians. Mixture models, like the previous example, commonly arise in market settings when sampling from a heterogeneous population (Sivan & Syrgkanis, 2013).

### 1.2. Related Works

We briefly outline three lines of relevant research for designing revenue-optimal mechanisms and testing the distributional assumptions of learning algorithms.

**Sample complexity of learning optimal auctions.** Cole & Roughgarden (2014) first proposed studying the number of samples needed to achieve a $(1 - \epsilon)$ multiplicative approximation to the optimal mechanism. Further work followed in single parameter settings (Huang et al. (2015) and Roughgarden & Schrijvers (2016), among others) culminating in the result of Guo et al. (2021) that provided tight bounds (up to poly-log factors) on the sample complexity of learning single item auctions. A related line of research has explored the robust learning of mechanisms from noisy or corrupt samples in both single-parameter and multi-parameter settings (Guo et al., 2021; Brustle et al., 2020; Cai & Daskalakis, 2017).

**Testable Learning.** The testable learning framework was first proposed by Rubinfeld & Vasilyan (2023) in the context of agnostically learning half-spaces over Gaussians and the uniform hypercube. Subsequent work by Gollakota et al. (2023b) gave testable learners for any concept class which admits low-degree sandwiching polynomials assuming the marginal is strongly log-concave. More recently, Gollakota et al. (2023a) showed testable learners for half-spaces for any marginal distribution that satisfies a Poincaire inequality. The testable learning framework has also been applied to settings that involve distribution shift and adversarial label noise (Diakonikolas et al., 2021; Klivans et al., 2024).

**Simple versus optimal auctions.** A fruitful line of research has explored whether revenue-optimal mechanisms can be well approximated by simpler mechanisms. Several important results in this line of research include Bulow & Klemperer (1996); Hartline & Roughgarden (2009); Alaei et al. (2019); Chawla et al. (2010); Haghpanah & Hartline (2015). A key assumption underlying many of these results is that the underlying distribution is regular. Lastly, Devanur et al. (2016) developed a key revenue monotonicity result that facilitates comparing the revenue of different mechanisms through the notion of first-order stochastic dominance.

## 2. Preliminaries

### 2.1. Auction Model

We focus on the fundamental single-item, $n$-bidder mechanism design setting. We assume each bidder $i$ draws their nonnegative valuation $v_i$ from an (unknown) distribution $D_i$ with a cumulative distribution function (cdf) of $F_i(x)$. For a given cdf $F(x)$, we define its (reverse) quantile function as $q(x) = 1 - F(x)$. In other words, $q(x)$ is the probability of sale if a item is posted at a price of $x$. We let $v = (v_1, v_2, ..., v_n)$ denote the vector of valuations which follows the product distribution $\mathbf{D} = \Pi_{i=1}^n D_i$. Henceforth, we utilize bold font to denote product distributions, which are typically (unless otherwise stated) over $n$ bidders. We assume the auctioneer only has sample access to $\mathbf{D}$ rather than explicit knowledge of the distribution function.

A mechanism $M$ consists of an allocation function $x$ and a payment function $p$. To run an auction, $M$ takes as input a vector of bids $b = (b_1, ..., b_n)$ and allocates the item according to $x(b) \in [0, 1]^n$, where $\sum_{i=1}^n x_i(b) \leq 1$, and charges bidders according to $p(b) \in \mathbb{R}^n$. For a vector of bids $b$, each bidder $i$ receives the item with probability $x_i(b)$ and pays $p_i(b)$. The utility of bidder $i$ is $u_i(x_i(b), p_i(b)) = v_i x_i(b) - p_i(b)$. The seller's goal is to find a mechanism that maximizes the expected value of the payments (i.e., maximizes $\mathbb{E}[\sum_{i=1}^n p_i(b)]$).

We restrict our attention to mechanisms which satisfy *Dominant Strategy Incentive Compatibility (DSIC)* and *Individual*

*Rationality (IR)*:

$$(DSIC) : u_i(v_i, b_{-i}) \geq u_i(b_i, b_{-i})$$

for all $i \in [n], v_i, b_i \in \mathbb{R}_{\geq 0}$ and $b_{-i} \in \mathbb{R}_{\geq 0}^{n-1}$,

$$(IR) : u_i(v_i, b_{-i}) \geq 0$$

for all $i \in [n], v_i \in \mathbb{R}_{\geq 0}$ and $b_{-i} \in \mathbb{R}_{\geq 0}^{n-1}$,

where $b_{-i}$ denotes the vector of bids from every agent except bidder $i$. The *DSIC* constraint ensures that a bidder maximizes their expected utility by truthfully bidding their value for an item, regardless of what other bidders do. The *IR* constraint ensures that a bidder's utility for participating in the auction is always nonnegative when they bid truthfully.

## 2.2. Revenue-Optimal Auctions

In this section, we outline a variety of useful theorems and definitions that we will refer to throughout the paper. A natural starting point is to define the expected revenue of a mechanism.

**Definition 1** (Expected Revenue of Mechanism). *The expected revenue of a mechanism $M$ on a distribution $\mathbf{D}$ is denoted by*

$$Rev(M, \mathbf{D}) = \mathop{\mathbb{E}}_{(v_1, \ldots, v_n) \sim \mathbf{D}} \left[ \sum_{i=1}^{n} p_i(v_i) \right].$$

*The Myerson optimal mechanism for $\mathbf{D}$, which we refer to as $M_{\mathbf{D}}$, maximizes this expectation and has expected revenue equal to $OPT(\mathbf{D})$.*

When important in context, we use the notation $Rev_n(M, D)$ to denote the revenue when a mechanism $M$ is run on $n$ i.i.d. bidders from the distribution $D$. In Myerson (1981), they introduce the fundamental notion of a regular distribution.

**Definition 2** (Regularity). *A distribution $D$ with cdf $F$ and density $f$ is called a regular distribution if its corresponding virtual value function $\phi(v) = v - \frac{1-F(v)}{f(v)}$ is monotonically non-decreasing for all $v \geq 0$. A product distribution $\mathbf{D}$ is regular if $D_i$ is regular for all $i \in [n]$.*

In Guo et al. (2021), they provide an equivalent characterization of regularity through their definition of a distribution's link function.

**Definition 3** (Link Function). *Let $D$ be a distribution with cdf $F(x)$. The link function of $D$ is defined as*

$$h(x; F) = \frac{1}{1 - F(x)}.$$

*We correspondingly denote the inverse of the link function as*

$$h^{-1}(x; h) = 1 - \frac{1}{h(x)}.$$

Before introducing the corresponding lemma, we recall the definition of first-order stochastic dominance.

**Definition 4** (First-Order Stochastic Dominance). *For two distributions $D$ and $D'$. We say that $D$ (first-order) stochastically dominates $D'$ (denoted by $D \succeq D'$) if*

$$F_D(x) \leq F_{D'}(x)$$

*for all $x \geq 0$. A product distribution $\mathbf{D}$ stochastically dominates another product distribution $\mathbf{D}'$ if $D_i \succeq D_i'$ for all $i \in [n]$.*

**Lemma 1** (Lemma 3.4 of Guo et al. (2021)). *A distribution $D$ is regular iff its link function $h(x; F)$ is a convex function of $x$. Moreover, for two distributions $D$ and $D'$, if $D$ first-order stochastically dominates $D'$ then $h(x; F) \leq h(x; F')$ for all $x \geq 0$.*

Later on, it will be useful to take the convex envelope of a non-convex link function in order to find a close regular distribution.

**Definition 5** (Convex Envelope). *The convex envelope of a function $f$ denoted by $Conv(f)$ is the maximum convex lower bound of $f$, i.e.,*

$$Conv(f) = \sup\{g(x) | \ g \text{ is convex}$$
$$\text{and } g(x) \leq f(x) \text{ for all } x \in \mathbb{R}_{\geq 0}\}.$$

*For a distribution $D$, we abuse notation by writing $Conv(D)$ to denote the convex envelope of the distribution's link function.*

Myerson (1981) fully characterizes the optimal single-item, multi-bidder mechanism, among all mechanisms that are *DISC* and *IR* compliant.

**Definition 6** (Myerson's Optimal Mechanism). *For a regular product distribution $\mathbf{D}$, the Myerson optimal mechanism $M_{\mathbf{D}}$ awards the item to the bidder $i$ with the highest non-negative virtual valuation (if such a bidder exists). The winner in turn pays $\phi_i^{-1}(\phi_0)$, where $\phi_0$ is the maximum of 0 and the second highest virtual value.*

Although Myerson's mechanism is optimal, it requires knowledge of the distribution function. Hence, simpler mechanisms such as the Vickrey auction and the anonymous price auction are also of interest to auction designers.

**Definition 7** (Vickrey Auction). *The Vickrey auction, denoted by $VA$, awards the item to the highest bidder and charges them a price equal to the second highest bid.*

**Definition 8** (Anonymous Price Auction). *The anonymous price auction with reserve $p^*$, denoted by $APA(p^*)$, awards the item to the highest bidder if their bid is at least $p^*$. The auction then charges them a price equal to the maximum of the second highest bid and $p^*$.*

## 2.3. Revenue Monotonicity

Often, we will need to compare the revenue of a mechanism on competing distributions. The work of Devanur et al. (2016) provides a convenient way to accomplish this through first-order stochastic dominance.

**Theorem 2** (Strong Revenue Monotonicity (Devanur et al., 2016))**.** *Given two product distributions* $\mathbf{D}$ *and* $\mathbf{D}'$ *such that* $\mathbf{D} \succeq \mathbf{D}'$*, it holds that*

$$Rev(M_{\mathbf{D}'}, \mathbf{D}) \geq Rev(M_{\mathbf{D}'}, \mathbf{D}') = OPT(\mathbf{D}').$$

## 2.4. Probability Metrics

To compare how close two distributions are, we will use Kolmogorov-Smirnov distance.

**Definition 9** (Kolmogorov-Smirnov Distance)**.** *For two distributions* $D$ *and* $D'$*, the Kolmogorov-Smirnov (KS) distance between* $D$ *and* $D'$ *is denoted by*

$$d_{KS}(D, D') = \sup_{x \in \mathbb{R}} |F_D(x) - F_{D'}(x)|.$$

*For a given distribution* $D$*, we denote the set of all distributions that are* $\alpha$*-close to* $D$ *in Kolmogorov-Smirnov distance by* $B_{KS,\alpha}(D) = \{D' : d_{KS}(D', D) \leq \alpha\}$*. A product distribution* $\mathbf{D}$ *is* $\alpha$*-close in Kolmogorov-Smirnov distance to another product distribution* $\mathbf{D}'$ *if* $d_{KS}(D'_i, D_i) \leq \alpha$ *for all* $i \in [n]$*.*

*For a Kolmogorov-Smirnov ball* $B_{KS,\alpha}(D)$*, we define the minimal regular distribution* $D^{min}$ *in this ball to have cdf* $F_{D^{min}}(x) = h^{-1}(x; \hat{h})$*, where*

$$\hat{h}(x) = \max_{\substack{D' \in B_{KS,\alpha}(D) \\ D' \, regular}} h(F_{D'}(x)) \quad \textit{for all } x \in \mathbb{R}_+.$$

In addition to KS distance, we will also utilize Kullback-Leibler Divergence to establish lower bounds on the minimum number of samples required to distinguish between two close distributions.

**Definition 10** (Kullback-Leibler Divergence)**.** *The Kullback-Leibler (KL) divergence between two distributions* $D$ *and* $D'$ *with pdfs* $f_D(v)$ *and* $f_{D'}(v)$ *is*

$$d_{KL}(D'||D) = \mathbb{E}_{v \sim D'} \left[ \ln \frac{f_{D'}(v)}{f_D(v)} \right].$$

*For a given distribution* $D$*, we denote the set of all distributions which are* $\alpha$*-close to* $D$ *in KL divergence by* $B_{KL,\alpha}(D) = \{D' : d_{KL}(D'||D) \leq \alpha\}$*. A product distribution* $\mathbf{D}'$ *is* $\alpha$*-close in KL divergence to another product distribution* $\mathbf{D}$ *if* $\sum_{i=1}^{n} d_{KL}(D'_i||D_i) \leq \alpha$*.*

Finally, for completeness, we also include a definition of the Dvoretzky–Kiefer–Wolfowitz inequality, which is used to bound the KS distance between an empirical distribution and a true distribution.

**Definition 11** (Dvoretzky–Kiefer–Wolfowitz (DKW) inequality (Dvoretzky et al., 1956; Massart, 1990))**.** *Given* $n$ *samples* $(X_i)_{i=1}^{n}$ *from a distribution* $D$*, the empirical distribution* $D_n$ *with cdf* $F_n(x) = n^{-1} \sum_{i=1}^{n} \mathbf{1}_{X_i \leq x}$ *satisfies the following inequality:*

$$Pr(d_{KS}(D_n, D) > \epsilon) \leq 2e^{-2n\epsilon^2}.$$

# 3. Regularized Testable Learning

As we explained above in many applications of auction theory, the regularity of the value distributions is assumed. Yet, it is not clear if we can test whether a given set of samples is drawn from a distribution that is regular. If such a test existed, it would allow us to know when we may apply certain results that require regularity to a particular source of data. In fact, in this section we show that we cannot efficiently test regularity directly, which motivates our regularized testable learning framework.

We first show in Section 3.1 that no meaningful result can be derived for revenue optimization using the original framework of Rubinfeld & Vasilyan (2023). We then present our new framework of regularized testable learning in Section 3.2, which we subsequently use to prove revenue guarantees in Section 5.

## 3.1. Testing Regularity for Revenue is Hard

To motivate Definition 14, we will show that it is impossible to design a tester-learner pair with nontrivial guarantees for revenue-maximizing auctions under the conventional learning and testing benchmarks defined below.

**Definition 12** (Conventional Learner)**.** *A family of distributions* $\mathcal{D}$ *has a* $f(\epsilon, \delta, n)$*-conventional learner* $\mathcal{L}$ *if given access to* $f(\epsilon, \delta, n)$ *i.i.d. samples from* $\mathbf{D} \in \mathcal{D}$*,* $\mathcal{L}$ *outputs a mechanism* $M$ *such that* $Rev(M, \mathbf{D}) \geq (1 - \epsilon)OPT(\mathbf{D})$ *with probability at least* $1 - \delta$*.*

**Definition 13** (Conventional Tester)**.** *An algorithm* $\mathcal{T}$ *is a* $h(\epsilon, \delta, n)$*-conventional tester for the class of regular distributions with respect to a learner* $\mathcal{L}$ *if it satisfies two constraints.*

- *Soundness: Suppose* $\mathcal{T}$ *is presented with* $h(\epsilon, \delta, n)$ *i.i.d. samples from a distribution* $\mathbf{D}$ *and outputs "Yes" with probability at least* $1 - \delta$*. Then, the conventional learning guarantee of Definition 12 should be achieved.*

- *Completeness: Suppose* $\mathcal{T}$ *is presented with* $h(\epsilon, \delta, n)$ *i.i.d. samples from a distribution* $\mathbf{D}$ *that is regular, then* $\mathcal{T}$ *should output "Yes" with probability at least* $1 - \delta$*.*

Under this definition, the learner must output a mechanism which achieves good revenue on the distribution relative

to its *optimal mechanism*. The crux of the issue with Definitions 12 and 13 is learning the optimal mechanism for an irregular distribution is generally infeasible. We demonstrate this in the following theorem by showing that we cannot simultaneously satisfy the soundness and completeness conditions present in Definition 13.

**Theorem 3.** *Let $\mathcal{L}$ be a $f(\epsilon, \delta, n)$-conventional learner. For any fixed number of samples $m = h(\alpha, \delta, n)$, there does not exist a $m$-conventional tester $\mathcal{T}$ for $\mathcal{L}$.*

*Proof.* Suppose for a contradiction that there exists a $m$-conventional tester $\mathcal{T}$. Without loss of generality, assume that $\delta = 1/4$. We begin by constructing a family of counter-examples parameterized by $\gamma$. Let $D_1$ be the standard uniform distribution over the interval $[0, 1]$ and let $D_2$ be a mixed distribution; with probability $1 - \gamma$, it is uniform over $[0, 1]$ and with probability $\gamma$ it is uniform over $[2^{1/\gamma}, 2^{1/\gamma} + 1]$. It is straightforward to verify that $D_1$ is regular and $D_2$ is irregular by examining the convexity of their respective link functions.

Without loss of generality, we will compare the distributions in terms of KL-divergence since $d_{KS}(D_1, D_2) = \gamma \leq -\ln(1 - \gamma) = d_{KL}(D_1 \| D_2)$. It is well-known (e.g., Huang et al. (2015)) that any algorithm which distinguishes between $D_1$ and $D_2$ with probability at least $3/4$ requires at least $d_{KL}(D_1 \| D_2)^{-1} = -\ln(1 - \gamma)^{-1}$ many samples. Myerson (1981) dictates that $M_{D_1}$ sets a reserve price of $1/2$ and $Rev(M_{D_1}, D_1) = 1/4$. It is simple to show that $M_{D_2}$ sets a reserve price of $2^{1/\gamma}$ and $Rev(M_{D_2}, D_2) = \gamma 2^{1/\gamma}$. Then,

$$\lim_{\gamma \to 0} \frac{Rev(M_{D_2}, D_2)}{Rev(M_{D_1}, D_1)} = \lim_{\gamma \to 0} 4\gamma 2^{1/\gamma} = \infty.$$

If $\mathcal{T}$ accepts $D_1$ with probability at least $3/4$, then $\mathcal{T}$ must accept $D_2$ with at least probability $3/4$ since for some $\gamma > 0$ the two distributions are indistinguishable with only $m$ samples. This violates soundness, as the revenue-maximizing auction for $D_2$ cannot be learned with $m$ samples. In particular, if the mechanism could be learned, then we could use the learner to distinguish between the two distributions. Alternatively, if $\mathcal{T}$ rejects $D_1$ with at least probability $1/4$, then completeness is violated, since $D_1$ is a regular distribution. □

### 3.2. Definition of the New Regularized Testable Learning Framework

We now formally present our framework of regularized testable learning for revenue-optimal auctions.

**Definition 14** (Regularized Learner). *The class of revenue-optimal mechanisms over regular distributions has an $(f(\alpha, \delta, n), g(\alpha, n))$-regularized learner $\mathcal{L}$ if given access*

*to $f(\alpha, \delta, n)$ i.i.d. samples from $\mathbf{D}$, $\mathcal{L}$ outputs a mechanism $M$ such that with probability $1 - \delta$*

$$Rev(M, \mathbf{D}) \geq (1 - g(\alpha, n))OPT(\mathbf{D}')$$

*for all regular distributions $\mathbf{D}'$ that are $\alpha$-close in KS distance to $\mathbf{D}$.*

**Definition 15** (Regularized Tester). *An algorithm $\mathcal{T}$ is a $h(\alpha, \delta, n)$-regularized tester for the class of regular distributions with respect to a learner $\mathcal{L}$ if it satisfies two constraints.*

- *Soundness: Suppose $\mathcal{T}$ is presented with $h(\alpha, \delta, n)$ i.i.d. samples from a distribution $\mathbf{D}$ and outputs "Yes" with probability at least $1 - \delta$. Then, the regularized learning guarantee of Definition 14 should be achieved.*

- *Completeness: Suppose $\mathcal{T}$ is presented with $h(\alpha, \delta, n)$ i.i.d. samples from a distribution $\mathbf{D}$ that is $\alpha$-close in Kolmogorov-Smirnov distance to a regular distribution, then $\mathcal{T}$ should output "Yes" with probability at least $1 - \delta$.*

Compared to the conventional learner, Definition 14 asks the regularized learner to produce a mechanism that generates strong revenue compared to the optimal mechanism of any nearby regular distribution. We correspondingly alter the completeness guarantee of Definition 13 to ensure that w.h.p. the regularized tester accepts any distribution that is sufficiently close to a regular distribution. This alternation allows us to prove meaningful revenue guarantees for unbounded, irregular distributions that are close to regular distributions. Although the above formulation is specific to the setting of learning revenue optimal auctions, in Appendix G we present a generic formulation of regularized testable learning that is applicable to a broader range of learning tasks.

## 4. A Regularized Tester for Revenue-Optimal Mechanisms

In this section, we present our regularized tester for revenue-optimal mechanisms and justify its function in relation to the learning guarantee of Definition 14. More precisely, any regularized learning algorithm should output a mechanism that performs well on the input distribution compared to the optimal revenue of any *close* regular distribution. Unfortunately, in Section 4.2 we show that there exist irregular distributions that may not be sufficiently close to any regular distribution. Hence, the role of our tester is to verify if there exists a close regular distribution to the input distribution.

## 4.1. The Testing Algorithm

Our regularized tester algorithm is similar in spirit to the main algorithms of Guo et al. (2019) and Guo et al. (2021) with an additional crucial conditional check at the end. Its goal is to determine if there exists a close regular distribution to the (possibly irregular) input distribution. Specifically, as input, the tester receives a distortion parameter $\alpha \in (0, 1)$ and $m$ samples from a product distribution $\mathbf{D} = \Pi_{i=1}^n D_i$. The algorithm then constructs a quantile-shifted version of the empirical distribution $\tilde{\mathbf{E}}$ that is both regular and stochastically dominated by true distribution $\mathbf{D}$ w.h.p.. The amount of quantile shift is determined via an application of the DKW inequality (Definition 11), along with a union bound since $\mathbf{D}$ is a product distribution. Unlike previous works, our tester then verifies w.h.p. if $\tilde{E}_i$ is in an $\alpha$-radius KS ball around $D_i$ for every $i \in [n]$. If so, the tester accepts; otherwise, it rejects.

---

**Algorithm 1** Regularized Tester

---

**Require:** $m$ i.i.d. samples from a distribution $\mathcal{D}$ and a distortion parameter $\alpha$
**Ensure: YES** or **NO**
1: Let $\mathbf{E} = \Pi_{i=1}^n E_i$ be the empirical distribution over the samples
2: **for** $i \leftarrow 1$ to $n$ **do**
3:     Construct $\hat{E}_i$ as follows:
4:     Let $q^{E_i}(v)$ be the quantile function of $E_i$
5:     Define

$$c = \sqrt{\frac{2\, q^{E_i}(v)\left(1 - q^{E_i}(v)\right)\ln\left(2mn\,\delta^{-1}\right)}{m}}$$
$$+ \frac{4\ln\left(2mn\,\delta^{-1}\right)}{m} + \alpha$$

6:     $q^{\hat{E}_i}(v) = \begin{cases} \max\left(0, q^{E_i}(v) - c\right) & \text{if } v > 0, \\ 1 & \text{if } v = 0 \end{cases}$

7:     $\tilde{E}_i = h^{-1}\left(\text{Conv}\left(\hat{E}_i(\cdot)\right)\right)$
8:     **for** each value $v$ in the support of $E_i$ **do**
9:         **if** $\left| q^{\tilde{E}_i}(v) - q^{E_i}(v) \right| > c$ **then**
10:             **return** NO
11:         **end if**
12:     **end for**
13: **end for**
14: **return** YES

---

A priori, it may not be clear whether the transformation associated with $q^{\hat{E}_i}(v)$ in Algorithm 1 corresponds to a valid distribution function. For completeness, we include a proof of this fact in Appendix B. This algorithm can be implemented in polynomial time (see Guo et al. (2021)). We will utilize this tester both in conjunction with a regularized learner (Section 5) and by itself (Sections 6 and 7).

## 4.2. Regular Distributions are not Dense

To motivate why we need to pair our regularized learner with Algorithm 1, we show that there exist distributions that are not suitably close to any regular distribution in terms of KS distance. We defer the proof of Lemmas 4 and 5 to Appendix A.

**Lemma 4** (Regular Distributions are not Dense). *There exists an irregular distribution $D$ and an $\alpha > 0$ such that for all regular distributions $D'$, it holds that $d_{KS}(D, D') > \alpha$.*

**Lemma 5** (Irregular Distributions are Dense). *For any distribution $D$ and $\alpha > 0$, there exists an irregular distribution $D'$ such that $D' \in B_{KS,\alpha}(D)$.*

## 5. Regularized Testable Learning for Optimal Auctions

The first setting that we apply our regularized tester to is learning revenue optimal auctions from i.i.d. samples, which we explore in this section. We begin by outlining our learning algorithm (Algorithm 2). Similar to the tester described in Section 3, the learner receives a distortion parameter $\alpha \in (0, 1)$ and $m$ samples from the product distribution $\mathbf{D} = \Pi_{i=1}^n D_i$. The learner then finds a regular distribution $\mathbf{E}'$ that is close to $\mathbf{D}$, whilst also having the property that $\mathbf{D} \succeq \mathbf{E}'$ and $\mathbf{D}' \succeq \mathbf{E}'$ for all regular distributions $\mathbf{D}'$ which are also close to $\mathbf{D}$. The learner then outputs the Myerson optimal mechanism with respect to $\mathbf{E}'$. As with Algorithm 1, the learner can be implemented in polynomial time.

We present a proof of the main theorem in Appendix C.

**Theorem 6.** *Let $\mathbf{D} = \Pi_{i=1}^n D_i$ be a product distribution, let $\alpha \in (0, 1)$ be a distortion parameter, and let $\delta > 0$. Additionally, fix $m = \tilde{\Omega}(\log(1/\delta)/\alpha^2)$. Suppose $\mathcal{T}$ is the tester described in Algorithm 1 and $\mathcal{L}$ is the learner described in Algorithm 2. Then, $\mathcal{T}$ is a $m$-sample regularized tester for the $(m, \sqrt{n}\alpha)$-regularized learner $\mathcal{L}$.*

## 6. A Tester for Bulow-Klemperer Theorem

In this section, we show that our regularized tester (Algorithm 1) can be used to verify when an approximate version of the Bulow-Klemperer theorem for irregular distributions holds. With i.i.d. regular bidders, the celebrated Bulow-Klemperer Theorem (Bulow & Klemperer, 1996) provides a simple method for generating at least as much revenue as the optimal mechanism; just recruit another bidder and run the Vickrey auction.

**Theorem 7** (Bulow-Klemperer). *Let $D$ be a regular distribution and $n$ a positive integer. Then,*

$$Rev_{n+1}(VA, D) \geq OPT_n(D),$$

*when the bidders are i.i.d. from $D$.*

**Algorithm 2** Regularized Learner

**Require:** $m$ i.i.d. samples from a distribution $\mathcal{D}$ and a distortion parameter $\alpha$

**Ensure:** Myerson's optimal auction $M_{\mathbf{E}'}$ w.r.t. $\mathbf{E}'$

1: Let $\mathbf{E} = \Pi_{i=1}^n E_i$ be the empirical distribution over the samples
2: **for** $i \leftarrow 1$ to $n$ **do**
3:     Construct $\hat{E}_i$ as follows:
4:     Let $q^{E_i}(v)$ be the quantile function of $E_i$
5:     Define

$$c = \sqrt{\frac{2\, q^{E_i}(v)\left(1 - q^{E_i}(v)\right)\ln\left(2mn\,\delta^{-1}\right)}{m}}$$
$$+\ \frac{4\ln\left(2mn\,\delta^{-1}\right)}{m}\ +\ \alpha$$

6:     $q^{\hat{E}_i}(v) = \begin{cases} \max\left(0,\, q^{E_i}(v) - c\right) & \text{if } v > 0, \\ 1 & \text{if } v = 0 \end{cases}$

7:     $\tilde{E}_i = h^{-1}\left(\text{Conv}\left(\hat{E}_i(\cdot)\right)\right)$
8:     Construct $E_i'$ as follows:
9:     $q^{E_i'}(v) = \begin{cases} \max(0, q^{\tilde{E}_i}(v) - \alpha) & \text{if } v > 0 \\ 1 & \text{if } v = 0 \end{cases}$

10: **end for**
11: Set $\mathbf{E}' = \Pi_{i=1}^n E_i'$
12: **return** Myerson's optimal auction $M_{\mathbf{E}'}$ w.r.t. $\mathbf{E}'$

In Theorem 8, we prove that if our tester approves an input distribution, then the Vickrey auction with an additional bidder achieves at least as much expected revenue on the distribution—modulo a multiplicative factor—as the optimal revenue of any close regular distribution. The proof is presented in Appendix D.

**Theorem 8.** *Let $D$ be a distribution, $\alpha \in (0,1)$ be a distortion parameter, and $\mathcal{T}$ be the tester described in Algorithm 1. Fix $m = \tilde{\Omega}(\log(1/\delta)/\alpha^2)$. With probability at least $1 - \delta$, if $\mathcal{T}$ outputs YES on input of $m$ i.i.d. samples from $\mathbf{D} = \Pi_{i=1}^n D$ and $\alpha$ then*

$$Rev_{n+1}(VA, D) \geq \left(1 - O\left(\sqrt{n\alpha}\right)\right) OPT_n(D')$$

*for any regular distribution $D'$ such that $D' \in B_{KS,\alpha}(D)$.*

## 7. A Tester for the Anonymous Price Auction

In multi-bidder settings where independence but not identical distributions is assumed, the optimal auction may implement different reserve prices for different bidders (Hartline & Roughgarden, 2009). A natural question is whether there exists a single anonymous reserve price for all bidders which achieves a constant fraction of the optimal revenue. The following theorem, originally shown by Hartline & Roughgarden (2009) and later improved by Alaei et al. (2019),

resolves this question for regular, non-identical distributions.

**Theorem 9.** *Let $\mathbf{D} = \Pi_{i=1}^n D_i$ be a product distribution, where $D_i$ is regular for all $i \in [n]$. There exists an anonymous reserve price $p^*$ such that*

$$Rev(APA(p^*), \mathbf{D}) \geq \frac{1}{e} OPT(\mathbf{D}).$$

We extend these results to irregular, non-identical distributions using the testing framework with the following theorem, whose proof is presented in Appendix E.

**Theorem 10.** *Let $\mathbf{D} = \Pi_{i=1}^n D_i$ be a product distribution, $\alpha \in (0,1)$ be a distortion parameter, and $\mathcal{T}$ be the tester described in Algorithm 1. Fix $m = \tilde{\Omega}(\log(1/\delta)/\alpha^2)$. With probability at least $1 - \delta$, if $\mathcal{T}$ outputs YES on input of $m$ i.i.d. samples from $\mathbf{D}$ and $\alpha$, then*

$$Rev(APA(p^*), \mathbf{D}) \geq \frac{1}{e}\left(1 - O\left(\sqrt{n\alpha}\right)\right) OPT(\mathbf{D}')$$

*for any regular product distribution $\mathbf{D}' = \Pi_{i=1}^n D_i'$ such that for all $i \in [n]$ it holds that $D_i' \in B_{KS,\alpha}(D_i)$.*

## 8. Information-Theoretic Lower Bound

In this section, we present an information-theoretic lower bound for our regularized testable learning framework. The single bidder lower bound involves finding two regular distributions which are $\alpha$-close in KS distance but for which no mechanism can simultaneously achieve more than a $(1 - \Omega(\sqrt{\alpha}))$ fraction of the optimal revenue for both distributions. We formalize this result in the lemma below, whose proof is found in Appendix F.

**Lemma 11.** *Let $M$ be some mechanism and let $\alpha < 0.1$. There exists an irregular distribution $D$ such that*

$$Rev(M, D) \leq (1 - \Omega(\sqrt{\alpha}))OPT(D')$$

*for some regular distribution $D' \in B_{KS,\alpha}(D)$.*

## 9. Conclusion

In this work, we develop our *Regularized Testable Learning* framework for mechanism design, wherein we relax the original testable learning benchmark proposed by Rubinfeld & Vasilyan (2023). This relaxation allows us to develop a tester to verify when we can expect to achieve a nontrivial fraction of the revenue of the optimal mechanism without assuming the samples come from a regular distribution. Unlike previous works, our tester-learner pair can provide meaningful revenue guarantees for irregular, unbounded distributions. We additionally show how our tester can be used to verify when important theorems from auction theory that rely on regularity can be safely applied.

For future work, we note three fruitful directions. First, it would be intriguing to see if one could develop a tester for our assumption that the distribution is a product distribution. Similarly, it would be interesting to understand if it is possible to develop a tester for the assumption that the bidders are i.i.d.. Finally, it would be interesting to establish a tight lower bound for the multi-bidder setting.

## Acknowledgements

We would like to thank Justin Dominic for insightful discussions on the empirical evaluation of our methods and the anonymous reviewers for their constructive feedback on improving the presentation of our results.

## Impact Statement

This paper presents work whose goal is to advance the field of testable learning in mechanism design. While there are societal consequences of our work, this paper is theoretical in nature. Hence, there are no particular societal consequences that we feel we must specifically highlight here.

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

# Appendix: Missing Proofs

## A. Proof of Lemmas 4 and 5

In this section, we prove two lemmas on the density of regular and irregular distributions. We begin by establishing a helpful claim.

**Claim 12.** *Let $D$ and $D'$ be two distributions with cdfs of $F_D$ and $F_{D'}$, respectively. Let $\alpha > 0$ and suppose $D \succeq D'$. If $h(x; F_D) > \frac{1}{1/h(x;F_{D'})-\alpha}$, then $F_D(x) - F_{D'}(x) > \alpha$.*

*Proof.*

$$
F_D(x) - F_{D'}(x) = (1 - F_{D'}(x)) - (1 - F_D(x))
$$
$$
= \frac{h(x; F_D) - h(x; F_{D'})}{h(x; F_D)h(x; F_{D'})}
$$
$$
= \frac{1}{h(x; F_{D'})} - \frac{1}{h(x; F_D)}
$$
$$
> \alpha
$$

$\square$

We can now prove that regular distributions are not dense.

**Lemma 4** (Regular Distributions are not Dense)**.** *There exists an irregular distribution $D$ and an $\alpha > 0$ such that for all regular distributions $D'$, it holds that $d_{KS}(D, D') > \alpha$.*

*Proof.* Let $D$ be the mixed distribution that is uniform over $[0, 1]$ with probability $1 - \epsilon$ and uniform over $[2^{1/\epsilon}, 2^{1/\epsilon} + 1]$ with probability $\epsilon$ for some $\epsilon < 0.1$. Additionally, let $\alpha = \epsilon/2$. Define

$$
D' = h^{-1}\left(\frac{1 - 4\alpha}{(3\alpha)(1 - \alpha)}x + \frac{1}{1 - \alpha}\right).
$$

By Lemma 1, $D'$ is regular since $h(x; F_{D'})$ is linear. Our goal will be to show that for $x \geq 1$ the link function of $D'$ lower bounds the link function of every regular distribution that is $\alpha$-close in Kolmogorov-Smirnov distance. Once this is established, we will use Claim 12 on $D'$ and $D$ to achieve a contradiction.

Let $\hat{D}$ be a regular distribution such that $d_{KS}(\hat{D}, D) \leq \alpha$. It holds that $h(0; F_{\hat{D}}) \leq h(0; F_{D'})$ and $h(1; F_{\hat{D}}) \geq h(1; F_{D'})$ since $F_{D'}(0) - F_D(0) = \alpha$ and $F_D(1) - F_{D'}(1) = \alpha$. Since every link function is a convex, increasing function, it must be that $h'(x; F_{\hat{D}}) \geq h'(x; F_{D'})$ for all $x \geq 1$. Moreover, we may utilize that fact that $h(1; F_{\hat{D}}) \geq h(1; F_{D'})$ to conclude that

$$
h(x; F_{\hat{D}}) \geq h(x; F_{D'})
$$

for all $x \geq 1$. We will now lower bound $h(x; F_{D'})$ and apply Claim 12. In particular, for $1 \leq x \leq 2^{1/\epsilon}$, $F_D(x) = 1 - 2\alpha$ so $h(x; F_D) = \frac{1}{2\alpha}$. Then,

$$
h(x; F_{D'}) = \frac{1 - 4\alpha}{(3\alpha)(1 - \alpha)}x + \frac{1}{1 - \alpha} > \frac{1}{\alpha} = \frac{1}{1/h(x; F_D) - \alpha}
$$

when $4 \leq x \leq 2^{1/\epsilon}$. By Claim 12, this implies $d_{KS}(\hat{D}, D) > \alpha$, which is a contradiction. $\square$

In contrast, we show below that irregular distributions are dense.

**Lemma 5** (Irregular Distributions are Dense)**.** *For any distribution $D$ and $\alpha > 0$, there exists an irregular distribution $D'$ such that $D' \in B_{KS,\alpha}(D)$.*

*Proof.* Without loss of generality, assume that $D$ is a regular distribution and fix $\alpha > 0$. Let $v_1$ be the value such that $1 - F_D(v_1) = \alpha$ and let $v_2$ be the value such that $1 - F_D(v_2) = \alpha/2$. We will construct an irregular distribution by shifting the $\alpha/2$ probability in the region $(v_1, v_2]$ to higher values. In particular, we may now define an irregular distribution $D'$ as follows:

$$F_{D'}(x) = \begin{cases} F_D(x), & 0 \le x \le v_1, \\ F_D(v_1), & v_1 < x \le v_2, \\ F_D((x - v_2) + v_1)), & v_2 < x. \end{cases}$$

It is easy to verify that $F_{D'}(x)$ is a valid cdf. We can also see that $D'$ is irregular, since $h(x; F_{D'})$ is not convex in the region $[0, v_2]$. We conclude the proof by observing that $d_{KS}(D', D) \le \alpha$ since we only shift around $\alpha/2$ amount of probability between the two distributions. □

## B. Proof of Lemma 13

For completeness, we include below a proof that the quantile shift described in Algorithms 1 and 2 yields a valid cumulative density function.

**Lemma 13.** *Let $D$ be a distribution with quantile function $q(v) = 1 - F(v)$ and let $\alpha \in (0, 1)$. Consider the distribution defined by the quantile function*

$$\bar{q}(v) = \begin{cases} \max(0, q(v) - c) & \text{if } v > 0, \\ 1 & \text{if } v = 0, \end{cases}$$

*where*

$$c = \sqrt{\frac{2q(v)(1 - q(v)) \ln(2mn\delta^{-1})}{m}} + \frac{4 \ln(2mn\delta^{-1})}{m} + \alpha.$$

*Then, it holds that $\bar{F}(v) = 1 - \bar{q}(v)$ is a valid cumulative distribution function.*

*Proof.* By construction, $\bar{F}(v)$ has the following properties: $\bar{F}(v) \in [0, 1]$, $\lim_{v \to -\infty} \bar{F}(v) = 0$, and $\lim_{v \to \infty} \bar{F}(v) = 1$. We focus on proving that $\bar{q}(v)$ is monotonically decreasing by taking the derivative with respect to $v$. Without loss of generality, we ignore additive constants and let $t = \frac{2 \ln(2mn\delta^{-1})}{m}$ be a positive constant, independent of $v$. Then,

$$\bar{q}(v) = 1 - F(v) - \sqrt{t \cdot (1 - F(v))F(v)}$$

and

$$\frac{d}{dv}\bar{q}(v) = -f(v) \cdot \left(1 + \frac{\sqrt{t}(1 - 2F(v))}{2\sqrt{(1 - F(v))F(v)}}\right).$$

Since $F(v)$ is non-decreasing, $-f(v)$ is non-positive. Therefore, we seek to verify that $\left(1 + \frac{\sqrt{t}(1 - 2F(v))}{2\sqrt{(1 - F(v))F(v)}}\right)$ is nonnegative from $[0, a]$ and nonpositive from $(a, v_{max}]$, where $a$ is a constant to be specified below and $v_{max}$ is the maximum value in the empirical distribution. The first condition will ensure that $\bar{F}(v)$ is non-decreasing for $v \in [0, a]$. The second condition will allow us to show that $\bar{F}(v) = 1$ for $v \in (a, v_{max}]$.

For the first condition, we rearrange the statement to prove instead that $\sqrt{t}(1 - 2F(v)) \ge -2\sqrt{(1 - F(v))F(v)}$. We can now break it into two cases.

- If $F(v) \le 1/2$, then this inequality is clearly true, as the RHS is always negative.

- If $F(v) \ge 1/2$, then we may square both sides and observe that we get the quadratic form $F(v)^2 - F(v) + \frac{t}{4(t+1)}$. This quadratic has two roots $(\frac{1}{2} \pm \frac{1}{2\sqrt{t+1}})$. The smaller root $v_1$ is vacuous and introduced by squaring operation. It is also covered under the first case since $v_1 < 1/2$. Consequently, we can determine that the derivative is nonpositive for $v \le \frac{1}{2} + \frac{1}{2\sqrt{t+1}}$.

Finally, for the region $v \in (\frac{1}{2} + \frac{1}{2\sqrt{t+1}}, v_{max}]$, we can show that $q(v) - c < 0$ using the fact that $q(v_{max}) - c < 0$ and $\frac{d}{dv}\bar{q}(v) \ge 0$ in this region. Thus, $\bar{q}(v) = 0$ for $v \in (\frac{1}{2} + \frac{1}{2\sqrt{t+1}}, v_{max}]$ as desired. □

## C. Proof of Upper Bound for Testable Learning

In this section, we will prove the main theorem for the testable learning of revenue-maximizing auctions. Before proving Theorem 6, we show a useful lemma.

**Lemma 14.** *Let $D$ be a regular distribution with cdf of $F_D(x)$ and $\alpha \in (0,1)$. Define the distribution $D'$ with the following cdf:*

$$F_{D'}(x) = \begin{cases} F_D(x) + \alpha, & 0 \leq x < F_D^{-1}(1-\alpha) \\ 1 & x \geq F_D^{-1}(1-\alpha). \end{cases}$$

*Then, $D'$ is also a regular distribution.*

*Proof.* By Lemma 1, $D'$ is regular if and only if its link function $h(x; F_{D'})$ is convex. We will take the derivative of $h(x; F_{D'})$ and show that it is an increasing function. In particular,

$$\begin{aligned} \frac{\partial h(x; F_{D'})}{\partial x} &= \frac{\partial}{\partial x} \left( \frac{1}{1 - F'_D(x)} \right) \\ &= \frac{\partial}{\partial x} \left( \frac{1}{1 - (F_D(x) + \alpha)} \right) \\ &= \frac{f_D(x)}{(1 - F_D(x) - \alpha)^2}. \end{aligned}$$

By the regularity of $D$, $\frac{f_D(x)}{(1-F_D(x))^2}$ is increasing, so $\frac{f_D(x)}{(1-F_D(x)-\alpha)^2}$ is also increasing. $\qquad\square$

We are now ready to present the proof of the main theorem.

**Theorem 6.** *Let $\mathbf{D} = \Pi_{i=1}^n D_i$ be a product distribution, let $\alpha \in (0,1)$ be a distortion parameter, and let $\delta > 0$. Additionally, fix $m = \tilde{\Omega}(\log{(1/\delta)}/\alpha^2)$. Suppose $\mathcal{T}$ is the tester described in Algorithm 1 and $\mathcal{L}$ is the learner described in Algorithm 2. Then, $\mathcal{T}$ is a $m$-sample regularized tester for the $(m, \sqrt{n}\alpha)$-regularized learner $\mathcal{L}$.*

*Proof.* We begin by showing that the tester's completeness guarantee holds with probability at least $1 - \delta$. Choose $m$ large enough so that

$$\sqrt{\frac{2q^{D_i}(v)(1 - q^{D_i}(v)) \ln{(2mn\delta^{-1})}}{m}} + \frac{4 \ln{(2mn\delta^{-1})}}{m} \leq \alpha$$

for every $i \in [n]$. Suppose that there exists a regular product distribution $\mathbf{D}'$ such that $D_i' \in B_{KS,\alpha}(D_i)$ for all $i \in [n]$. By Lemma 5 of Guo et al. (2019), with probability $1 - \delta$ for every value $v$ and every $i \in [n]$, it holds that

$$|q^{E_i}(v) - q^{D_i}(v)| \leq \sqrt{\frac{2q^{D_i}(v)(1 - q^{D_i}(v)) \ln{(2mn\delta^{-1})}}{m}} + \frac{4 \ln{(2mn\delta^{-1})}}{m} \leq \alpha.$$

Fix an index $i \in [n]$. To show the correctness of the tester, we will prove that $|q^{\tilde{E}_i} - q^{E_i}| \leq 2\alpha$. Observe that by construction $D_i' \succeq \tilde{E}_i \succeq \hat{E}_i$. Hence, by a sandwiching argument, we only need to upper bound the quantile distance between $D_i'$ and $E_i$ and between $\hat{E}_i$ and $E_i$, respectively. Specifically,

$$|q^{D_i'}(v) - q^{E_i}(v)| \leq |q^{D_i'}(v) - q^{D_i}(v)| + |q^{D_i}(v) - q^{E_i}(v)| \leq 2\alpha$$

and

$$|q^{\hat{E}_i}(v) - q^{E_i}(v)| \leq 2\alpha.$$

Thus, the tester outputs YES with probability at least $1 - \delta$.

We now prove the tester's soundness guarantee. Suppose that $\mathcal{T}(\mathbf{D}, \alpha, m)$ outputs $YES$ and let $M$ be the mechanism outputted by $\mathcal{L}(\mathbf{D}, \alpha, m)$ (that is, $Rev(M, \mathbf{E}') = OPT(\mathbf{E}')$). As before, we begin by noting that $D_i' \succeq \tilde{E}_i \succeq E_i'$ and that

$D_i \succeq E_i'$ by construction. Furthermore, by Lemma 14, $E'$ is regular. We now need to bound the distance between $E_i'$ and any regular distribution $D_i' \in B_{KS,\alpha}(D_i)$ for all $i \in [n]$. We accomplish this through the triangle inequality,

$$|q^{E_i'}(v) - q^{D_i'}(v)| \le |q^{E_i'}(v) - q^{\tilde{E}_i}(v)| + |q^{\tilde{E}_i}(v) - q^{E_i}(v)| + |q^{E_i}(v) - q^{D_i}(v)| + |q^{D_i}(v) - q^{D_i'}(v)|$$
$$\le \alpha + 2\alpha + \alpha + \alpha$$
$$\le 5\alpha.$$

With this bound established, using Theorem 4.3 of Guo et al. (2021), with probability at least $1 - \delta$ it holds that

$$OPT(\mathbf{E}') \ge \left(1 - O\left(\sqrt{n\alpha}\right)\right) OPT(\mathbf{D}'),$$

for any regular product distribution $\mathbf{D}' = \Pi_{i=1}^n D_i'$ such that for all $i \in [n]$ it holds that $D_i' \in B_{KS,\alpha}(D_i)$. To complete the proof, we utilize the fact that $\mathbf{D} \succeq \mathbf{E}'$ and apply Theorem 2 to conclude that

$$Rev(M, \mathbf{D}) \ge OPT(\mathbf{E}') \ge \left(1 - O\left(\sqrt{n\alpha}\right)\right) OPT(\mathbf{D}').$$

$\square$

## D. Proof of Tester for Bulow-Klemperer

In this section, we prove that our tester can be used to verify when we may apply the Bulow-Klemperer Theorem to a data source. We begin by establishing a helpful revenue monotonicity claim.

**Claim 15.** *Let $D, D'$ be two distributions such that $D' \succeq D$ and the bidders for each distribution are i.i.d., respectively. Then, it holds that*

$$Rev_{n+1}(VA, D') \ge Rev_{n+1}(VA, D).$$

*Proof.* Recall that the Vickrey auction charges the winning bidder the value of the second highest bid. Hence, $Rev_{n+1}(VA, D)$ is equivalent to the expected value of the $n$th order statistic for the distribution $D$. A well-known fact is that the cdf of the nth order statistic is

$$G(F_D(x)) = (n+1)F_D(x)^n(1 - F_D(x)) + F_D(x)^{n+1}.$$

A cumulative density function by definition is monotonically non-decreasing. Thus, since $F_D(x) \ge F_{D'}(x)$, this implies that $G(F_D(x)) \ge G(F_{D'}(x))$. Stochastic dominance further implies that the expected value of $n$th order statistic for the distribution $D'$ is at least as large as the expected value of the $n$th order statistic of the distribution $D$. $\square$

In the following, we establish the main theorem of this section.

**Theorem 8.** *Let $D$ be a distribution, $\alpha \in (0,1)$ be a distortion parameter, and $\mathcal{T}$ be the tester described in Algorithm 1. Fix $m = \tilde{\Omega}(\log(1/\delta)/\alpha^2)$. With probability at least $1 - \delta$, if $\mathcal{T}$ outputs YES on input of $m$ i.i.d. samples from $\mathbf{D} = \Pi_{i=1}^n D$ and $\alpha$ then*

$$Rev_{n+1}(VA, D) \ge \left(1 - O\left(\sqrt{n\alpha}\right)\right) OPT_n(D')$$

*for any regular distribution $D'$ such that $D' \in B_{KS,\alpha}(D)$.*

*Proof.* Suppose $T(\mathbf{D}, \alpha, m)$ outputs YES. Then with probability at least $1 - \delta$, there exists a regular distribution $D'$ such that $D' \in B_{KS,\alpha}(D)$. Let $D^{min}$ be the minimal regular distribution in $B_{KS,\beta}(D)$ and define $\tilde{D}$ to have the cdf:

$$F_{\tilde{D}}(x) = \begin{cases} F_{D^{min}}(x) + \alpha, & 0 \le x \le F_{D^{min}}^{-1}(1-\alpha) \\ 1 & x \ge F_{D^{min}}^{-1}(1-\alpha). \end{cases} \tag{*}$$

Notice, $D \succeq \tilde{D}$ and $d_{KS}(\tilde{D}, D) \le 2\alpha$. Additionally, by Lemma 14, $\tilde{D}$ is regular and $D' \succeq \tilde{D}$ for all regular distributions $D' \in B_{KS,\beta}(D)$. We complete the proof through the following chain of inequalities:

$$\begin{aligned} \left(1 - O\left(\sqrt{n\alpha}\right)\right) OPT_n(D') &\le OPT_n(\tilde{D}) && \text{Theorem 4.3 (Guo et al., 2021)} \\ &\le Rev_{n+1}(VA, D_m') && \text{Theorem 7} \\ &\le Rev_{n+1}(VA, D) && \text{Claim 15} \end{aligned}$$

for all regular distributions $D' \in B_{KS,\alpha}(D)$. $\square$

# E. Proof of Tester for Anonymous Price Auction

Below, we show that our tester can be used to verify when the anonymous price auction achieves a non-trivial fraction of the optimal revenue of any close regular distribution.

**Theorem 10.** *Let $\mathbf{D} = \Pi_{i=1}^n D_i$ be a product distribution, $\alpha \in (0, 1)$ be a distortion parameter, and $\mathcal{T}$ be the tester described in Algorithm 1. Fix $m = \tilde{\Omega}(\log(1/\delta)/\alpha^2)$. With probability at least $1 - \delta$, if $\mathcal{T}$ outputs YES on input of $m$ i.i.d. samples from $\mathbf{D}$ and $\alpha$, then*

$$Rev(APA(p^*), \mathbf{D}) \geq \frac{1}{e} \left(1 - O\left(\sqrt{n}\alpha\right)\right) OPT(\mathbf{D}')$$

*for any regular product distribution $\mathbf{D}' = \Pi_{i=1}^n D_i'$ such that for all $i \in [n]$ it holds that $D_i' \in B_{KS,\alpha}(D_i)$.*

*Proof.* Let $\mathbf{D}' = \Pi_{i=1}^n D_i'$ be a regular product distribution such that for all $i \in [n]$ it holds that $D_i' \in B_{KS,\alpha}(D_i)$. Further, let $\tilde{\mathbf{D}}$ be the regular distribution defined in Appendix D by $(*)$. Then,

$$\frac{1}{e} \left(1 - O\left(\sqrt{n}\alpha\right)\right) OPT(\mathbf{D}') \leq \frac{1}{e} OPT(\tilde{\mathbf{D}}) \qquad \text{Theorem 4.3 (Guo et al., 2021)}$$

$$\leq Rev(APA(p^*), \tilde{\mathbf{D}}). \qquad \text{Theorem 9}$$

For the remainder of the proof, we will focus on showing that

$$Rev(APA(p^*), \tilde{\mathbf{D}}) \leq Rev(APA(p^*), \mathbf{D})$$

utilizing the property that $\mathbf{D} \succeq \tilde{\mathbf{D}}$. Let $X_{(i)}$ denote the random variable of the $i$th order statistic from an $n$-sample. By the definition of $APA(p^*)$,

$$Rev(APA(p^*), \tilde{\mathbf{D}}) = Pr_{\mathbf{D}}(X_{(n-1)} > p^*) \cdot \mathbb{E}_{\mathbf{D}}[X_{(n-1)}|X_{(n-1)} > p^*]$$
$$+ Pr_{\mathbf{D}}(X_{(n)} \geq p^* \cap X_{(n-1)} \leq p^*) \cdot p^* + Pr_{\mathbf{D}}(X_{(n)} < p^*) \cdot 0.$$

It is straightforward to establish that

$$x_1^{\mathbf{D}} := Pr_{\mathbf{D}}(X_{(n)} < p^*) = \Pi_{i=1}^n F_i(p^*),$$

$$x_2^{\mathbf{D}} := Pr_{\mathbf{D}}(X_{(n)} \geq p^* \cap X_{(n-1)} \leq p^*) = \sum_{i=1}^n (1 - F_i(p^*)) \cdot \Pi_{j \neq i} F_j(p^*),$$

$$x_3^{\mathbf{D}} := Pr_{\mathbf{D}}(X_{(n-1)} > p^*) = 1 - \sum_{i=1}^n ((1 - F_i(p^*)) \cdot \Pi_{j \neq i} F_j(p^*)) - \Pi_{i=1}^n F_i(p^*).$$

First, $x_1^{\tilde{\mathbf{D}}} \leq x_1^{\mathbf{D}}$ because $F_i^{\tilde{\mathbf{D}}}(p^*) \geq F_i^{\mathbf{D}}(p^*)$ for all $i \in [n]$. Then, using the fact that $x_1^{\mathbf{D}} + x_2^{\mathbf{D}} + x_3^{\mathbf{D}} = x_1^{\tilde{\mathbf{D}}} + x_2^{\tilde{\mathbf{D}}} + x_3^{\tilde{\mathbf{D}}} = 1$, it holds that

$$p^*(x_2^{\tilde{\mathbf{D}}} + x_3^{\tilde{\mathbf{D}}}) \leq p^*(x_2^{\mathbf{D}} + x_3^{\mathbf{D}}).$$

To complete the proof, we need to establish that $\mathbb{E}_{\mathbf{D}}[X_{(n-1)}|X_{(n-1)} > p^*] \geq \mathbb{E}_{\tilde{\mathbf{D}}}[X_{(n-1)}|X_{(n-1)} > p^*]$. We will establish this by showing that $x_3^{\mathbf{D}} \geq x_3^{\tilde{\mathbf{D}}}$, which implies that the cdf of the $X_{(n-1)}$ under $\mathbf{D}$ stochastically dominates the cdf of the $X_{(n-1)}$ under $\tilde{\mathbf{D}}$; consequently, the dominating distribution has a larger conditional expected value.

To accomplish this, we will show that the derivative of $x_3^{\mathbf{D}}$ with respect to $F_k$ for every $k \in [n]$ is non-positive. In particular,

$$\frac{\partial}{\partial F_k} \left(1 - \sum_{i=1}^n ((1 - F_i(x)) \cdot \Pi_{j \neq i}(F_j(x)) - \Pi_{i=1}^n F_i(x)\right)$$

$$= \frac{\partial}{\partial F_k} \left(1 - \sum_{i=1}^n \left(\Pi_{j \neq i} F_j(x) - \Pi_{j=1}^n F_j(x)\right) - \Pi_{i=1}^n F_i(x)\right)$$

$$= -f_k \cdot \left((n-1)\Pi_{i \neq k} F_i(x) - \sum_{i \neq k} \Pi_{j \neq i} F_j(x)\right)$$

$$\leq 0$$

since $(n-1)\Pi_{i\neq k}F_i(x) \geq \sum_{i\neq k}\Pi_{j\neq i}F_j(x)$ and $f_k \geq 0$. To finish the proof, we note that

$$
\begin{aligned}
Rev(APA(p^*), \tilde{\mathbf{D}}) &= p^* \cdot x_2^{\tilde{\mathbf{D}}} + \mathbb{E}_{\tilde{\mathbf{D}}}[X_{(n-1)}|X_{(n-1)} > p^*] \cdot x_3^{\tilde{\mathbf{D}}} \\
&\leq p^* \cdot x_2^{\mathbf{D}} + \mathbb{E}_{\mathbf{D}}[X_{(n-1)}|X_{(n-1)} > p^*] \cdot x_3^{\mathbf{D}} \\
&= Rev(APA(p^*), \mathbf{D}).
\end{aligned}
$$

$\square$

## F. Proof of Lower Bound for Testable Learning

In this section, we present a lower bound proof for the testable learning of revenue-maximizing auctions in the single-bidder setting.

**Lemma 11.** *Let $M$ be some mechanism and let $\alpha < 0.1$. There exists an irregular distribution $D$ such that*

$$
Rev(M, D) \leq (1 - \Omega(\sqrt{\alpha}))OPT(D')
$$

*for some regular distribution $D' \in B_{KS,\alpha}(D)$.*

*Proof.* Let $\beta > 0$ be a value to be defined later. We utilize the same two regular distributions $D_1, D_2$ which appear in Huang et al. (2015) and have the following cdfs:

$$
F_1(x) = 1 - \frac{1}{x+1},
$$

$$
F_2(x) = \begin{cases} 1 - \frac{1}{x+1}, & 0 \leq x \leq \frac{1-\beta}{\beta} \\ 1 - \frac{\beta^2}{x-(1-\beta)} & x > \frac{1-\beta}{\beta}, \end{cases}
$$

respectively. By Lemma 4.10 and Lemma 4.11 of Huang et al. (2015), there cannot exist a mechanism $M$ that simultaneously achieves a $1-\beta/2$ fraction of the optimal revenue for both $D_1$ and $D_2$. Wlog, suppose $Rev(M, D_1) \leq (1-\beta/2)OPT(D_1)$. By Lemma 5, there must exist an irregular distribution $D$ such that $D_1 \succeq D$ and $D_1, D_2 \in B_{KS,\beta^2}(D)$. We may then apply strong revenue monotonicity to find

$$
Rev(M, D) \leq (1 - \beta/2)OPT(D_1).
$$

To complete the proof, we bound the Kolmogorov distance between $D_1$ and $D_2$. By solving for the derivative's zero, it can be shown that

$$
\max |F_2(x) - F_1(x)| = \frac{\beta^2}{2 - \beta} \leq \beta^2.
$$

We achieve the desired bound by setting $\beta = \sqrt{\alpha}$. $\square$

## G. Generic Regularized Learning Definition

Below we present a generic version of our regularized testable learning framework, which we believe will be applicable to a variety of settings outside revenue-maximizing auctions.

**Definition 16** (Generic Regularized Learner). *An algorithm $\mathcal{L}$ is a $(f(\alpha, \delta, n), g(\alpha, n))$-sample regularized learner for the concept class $\mathcal{C}$ with respect to a family of distributions $\mathcal{D}$ and a hypothesis class $\mathcal{H}$ if given access to $f(\alpha, \delta, n)$ i.i.d. samples from a $n$-dimensional distribution $\mathbf{D}$, $\mathcal{L}(\mathbf{D}, \alpha, m)$ outputs a hypothesis $h \in \mathcal{H}$ such that with probability at least $1 - \delta$*

$$
h(\mathbf{D}) \geq (1 - g(\alpha, n))c^*(\mathbf{D}')
$$

*for all $\alpha$-close distributions $\mathbf{D}' \in \mathcal{D}$ to $\mathbf{D}$, where $c^* \in \mathcal{C}$ is the optimal concept with respect to $\mathbf{D}'$.*

**Definition 17** (Generic Regularized Tester). *An algorithm $\mathcal{T}$ is a $f(\alpha, \delta, n)$-sample tester for a class of distributions $\mathcal{D}$ with respect to a learner $\mathcal{L}$ if it satisfies two constraints.*

- *Soundness: Suppose $\mathcal{T}$ is presented with $f(\alpha, \delta, n)$ i.i.d. samples from a distribution $\mathbf{D}$ and outputs "Yes" with probability at least $1 - \delta$. Then, the regularized learning guarantee of Definition 16 should be achieved.*

- *Completeness: Suppose the $\mathcal{T}$ is presented with $f(\alpha, \delta, n)$ i.i.d. samples samples from a distribution $\mathbf{D}$ that is $\alpha$-close to some distribution $\mathbf{D}' \in \mathcal{D}$, then $\mathcal{T}$ should output "Yes" with probability at least $1 - \delta$.*

In the case of learning revenue-optimal mechanisms, $\mathcal{C}$ is the set of revenue-optimal *DISC* and *IR* compatible mechanisms, $\mathcal{H}$ is the set of all *DISC* and *IR* compatible mechanisms, and $\mathcal{D}$ is the family of regular distributions.

