# OpenReview forum: "Safely Learning Optimal Auctions: A Testable Learning Framework for Mechanism Design"
_ICML.cc/2025/Conference — ICML 2025 poster_

### Official Review · Reviewer_VbU2 · 2025-03-12

**Overall Recommendation:** 3

**Summary:**

The authors study a variant of Rubinfeld and Vasilyan's *testable learning* in mechanism design, and give a concrete tester-learner for basic auction settings.

Many classical results in mechanism design, e.g. Myerson's Optimal Mechanism for auctions, require the underlying distribution over valuations be *regular*, a specific technical condition requiring $v-\frac{1-F(v)}{f(v)}$ be non-decreasing. The authors attempt to develop a "tester-learner" for this problem, meaning that given a bounded number of samples from the distribution, one should either

1) detect the distribution is irregular and output **FAIL**, or
2) output a (near)-optimal mechanism

allowing the user to safely distinguish whether it is possible to use the algorithm on certain data from an unknown distribution. Unfortunately, for revenue maximization, the authors show the above guarantee for testing regularity is impossible. Namely they observe that it is easy to construct statistically indistinguishable distributions with far optimal revenue

Motivated in this context, the authors introduce a relaxed version of testable learning for mechanism design which, given an unknown distribution D', only requires the learner to compete with the best regular distribution near D' in Kolmogorov-Smirnov distance. They give a tester-learner in this context for revenue maximization over independent bidders as well as for the Bulow-Klemperer Theorem and anonymous price auctions (i.e. that adding a bidder can get optimal revenue, and that a fixed reserve price across bidders can achieve within a constant of optimal).

At a technical level, the authors algorithm works by constructing a regular distribution from the convex envelope of a quantile-shifted empirical estimate. They reject if this distribution is far from the original, and otherwise argue since it is stochastically dominated by the original distribution, applying Myerson's optimal mechanism to this shifted distribution gives the desired guarantee.

**Claims And Evidence:**

Yes

**Essential References Not Discussed:**

N/A

**Experimental Designs Or Analyses:**

N/A

**Methods And Evaluation Criteria:**

Yes

**Other Comments Or Suggestions:**

Def 13: It reads like D is known now since it is not drawn from some class as in the previous def, but presumably this is not what is meant?

Typos: “It’s goal”, “minium”

**Other Strengths And Weaknesses:**

Removing the need for strong distributional assumptions is always a welcome direction in any type of learning, mechanism design included, and the tester-learner framework is an interesting approach to this problem. The authors give extensions in this sense to several classical theorems, potentially broadening their use case in theory and practice.

The main weakness of the work is that the relaxed version of completeness seems too weak to accomplish what the authors set out to do. Since the optimality guarantee is with respect to nearby regular distributions (and not the original distribution itself) and there are irregular distributions with *no* nearby such distributions, this means there are underlying distribution on which the full "tester-learner" fails, i.e. it may output yes and an arbitrarily bad mechanism. The whole point of Rubinfeld-Vasilyan's idea is to avoid this type of scenario, and ensure one can always trust the output of the tester-learner pair. Presumably there may also be settings where the optimum value of nearby regular distributions is simply much lower, which is better than the prior issue, but still seems somewhat questionable as a "success case" of the learner. I would suggest the authors spend a bit more space justifying why their relaxation and tester-learners are useful in this context (beyond just the fact that one cannot use standard completeness guarantee, which is not imo of itself a sufficient motivation for the proposed relaxed version).

EDIT: The authors have partially addressed my concerns regarding the completeness of their tester-learner, though the issue of nearby regular distributions with significantly worse value remains. I have increased my score accordingly.

**Questions For Authors:**

See above: can you provide some clarification why the tester-learner framework remains useful in a setting where there are still distributions over which one cannot trust its output? This seems antithetical to the main point of tester-learners as introduced in Rubinfeld-Vasilyan (and indeed as the authors present as a way to *know* when one can safely apply a mechanism!)

**Relation To Broader Scientific Literature:**

The key contribution of this paper is to give the first variants of several classic results in mechanism design with non-trivial guarantees for irregular distributions. Prior results rely strongly on regularity, while the authors show that at least for irregular distributions with some nearby regular distribution, it is either possible to detect irregularity or to output a mechanism as good as on the nearby distribution.

**Theoretical Claims:**

I checked the lower bound proofs which seem correct, and don't see any issues with the other claims.

---

> ### Author Rebuttal · Authors · 2025-04-01
>
> Thank you for carefully reading our paper. We appreciate the constructive feedback and comments.
>
> We would like to address your concern that our relaxed version of the completeness guarantee is too weak for what we wish to show. We agree that the completeness condition as mistakenly written in a Definition 14 is too weak. We should have instead written: “Suppose the tester T is presented with $h(\alpha,\delta,n)$ i.i.d. samples from a distribution $D$. $T$ should output yes with probability at least $1-\delta$ if and only if there exists a regular distribution that is $\alpha$-close to $D$. This stronger completeness guarantee avoids the scenario where the tester outputs yes on a distribution that has no close regular distribution, leading to a bad learning outcome. Importantly, in the proof of Theorem 6, we show that if there is a close regular distribution to the input distribution then the tester will find it and output yes.
>
> The other direction is also simple to show. Namely, the Tester operates by explicitly constructing a regular distribution and testing whether this distribution is close to the input distribution. If the regular distribution that the Tester constructs is far away from the input distribution, then we can be sure that there is no close regular distribution. More specifically, in the Testing algorithm we first construct a (likely irregular) surrogate distribution $\hat{E}$ that is close to the input distribution and whose link function upperbounds the link function of any close regular distribution. We then take the convex envelope of this link function, which yields a new link function that is the greatest convex lowerbound. Since convex link functions (Lemma 1) correspond to regular distributions, this procedure has located a regular distribution. If this regular distribution is not close to the input distribution, then by properties of the convex envelope and link function, we can be sure that there does not exist a close regular distribution. This update to the completeness condition is straightforward and does affect any of our theoretical results. We are committed to adding more details to clarify this in the next revision of the manuscript.

---

### Official Review · Reviewer_kyL3 · 2025-03-15

**Overall Recommendation:** 2

**Summary:**

Summary:

The paper proposes a framework for testably learning revenue-optimal auctions. In this setting, an auction designer has access to m samples (bids) and aims to design a DSIC and IR auction that maximizes revenue. Unlike prior work, which typically assumes conditions like regularity or MHR without testing their validity, the authors propose a framework to actively test for regularity. The goal is for the test to output YES with high probability when the distribution is regular, and for the learned auction to achieve high revenue (competitive with the optimal revenue of all close regular distributions) when it does so.

While the paper addresses a relevant problem, the technical novelty and significance of the work are unclear, and the presentation requires improvement to meet the standards for ICML.

Detailed comments:

1. The presentation needs improvement to enhance clarity and readability. Several terms are used without proper definition or are introduced too late. Examples include:
   - Stochastic dominance in Lemma 1 is mentioned before being defined.
   - The term "m-regularized tester" in Theorem 3 is not clearly defined.
   - In Definition 9, it is not initially clear that m refers to the number of samples.
   - The notation m is confusing, as it initially suggests samples from the product distribution rather than individual distributions.

These issues are individually minor but collectively hinder the readability of the paper.

2. The technical novelty of the work is not well-explained. The authors do not adequately differentiate their contributions and proof techniques from prior work.
   - For example, Algorithm 2 appears to be largely similar to Guo et al. (2019), while Algorithm 1 only adds a testing condition.
   - Clearly and explicitly stating the novel aspects of the work would strengthen the paper.
   - The purpose and significance of Section 6 are unclear and feel somewhat like an afterthought.

Overall, while the paper addresses an interesting problem, it needs better presentation and a clearer articulation of its technical contributions to be suitable for publication.

**Claims And Evidence:**

See above.

**Essential References Not Discussed:**

See above.

**Experimental Designs Or Analyses:**

See above.

**Methods And Evaluation Criteria:**

See above.

**Other Comments Or Suggestions:**

See above.

**Other Strengths And Weaknesses:**

See above.

**Questions For Authors:**

See above.

**Relation To Broader Scientific Literature:**

See above.

**Theoretical Claims:**

See above.

---

> ### Author Rebuttal · Authors · 2025-04-01
>
> Thank you for carefully reading our paper and providing helpful feedback and comments. We would first like to address your presentation concerns. We are committed to improving the preliminaries by ensuring that the definitions are introduced in the proper order. We will also improve the “m notation” to clarify when the samples are drawn from a product distribution versus an individual distribution. We believe these changes you suggest, among others, can be completed promptly.
>
> We would also like to address the significance of our work in comparison to [1,2].
>
> **Comparison with [1,2].** Both [1,2] only provide results for regular or bounded distributions. We demonstrate that without these assumptions it is necessary to relax the conventional learning benchmark found in the existing literature. Consequently, our learning benchmark is also different than [1,2].
>
> **Technical contribution.** It is indeed the case that many of the techniques that we use are based on proof techniques from [1,2], but (1) this is a coincidence, the problem that we are exploring is important on its own, and it happens that the techniques from [1, 2] are useful, and more importantly, (2) we cannot use the techniques from [1,2] in a black-box way. Instead, we need to carefully combine these techniques using some novel ideas. We believe that the fact that the techniques are based on [1,2] should not subtract value from our paper because the problem that we introduce and solve is well-motivated and important.
>
> Additionally, both our tester and learner differ from [1,2]. In particular, for our tester, we introduce a new conditional check on the difference between the empirical quantiles and the quantiles of a close regular distribution. If this difference is too large, it indicates that there does not exist a regular distribution within the $\alpha$ KL ball around our target distribution and thus we cannot expect our learner’s guarantee to hold. Our learner also differs from [1,2] in that the algorithm must further adjust the quantiles of the close regular distribution it finds to ensure that it is stochastically dominated by our target distribution.
>
> Finally, the primary purpose of Section 6 (A Tester for the Bulow-Klemperer Theorem) is to demonstrate that our Tester algorithm has utility outside of just being directly paired with a learning algorithm; the tester can be used to verify when important theorems in auction theory that rely on regularity can be expected to apply to a dataset. In particular, we demonstrate that we can test when the fundamental Bulow-Klemperer theorem can be applied to a dataset. This theorem says that when a distribution is regular and the bidders are i.i.d., there exists a simple method to generate as much revenue as the optimal mechanism. Namely, the auctioneer should just recruit another bidder and run the Vickrey Auction.
>
>
> [1] Chenghao Guo, Zhiyi Huang, and Xinzhi Zhang. "Settling the sample complexity of single-parameter revenue maximization." Proceedings of the 51st Annual ACM SIGACT Symposium on Theory of Computing (pp. 662-673) (2019).
>
> [2] Wenshuo Guo, Michael Jordan, and Emmanouil Zampetakis. "Robust learning of optimal auctions." Advances in Neural Information Processing Systems 34 (2021).

---

### Official Review · Reviewer_CS1L · 2025-03-20

**Overall Recommendation:** 3

**Summary:**

This paper considers auctions with possibly regular or near regular distributions, and considers a two step process of a) testing for regularity, and b) designing an approximately optimal auction if the distribution tests as near regular. Regularity is a key distributional assumption in auction theory: it states that the revenue from setting a price for a single agent is convex, so e.g., you could use local search to learn an optimal reserve. And, setting that optimal reserve in a Vickrey auction gives you the optimal auction.

Prior work has considered similar questions, how to test distributions, how many samples are needed to learn almost the optimal auction, how much error can be present in draws from a regular distribution, and bounds for samples needed to learn near optimal auctions in possibly irregular settings (without testing).

This work focuses on empirical draws from a distribution that could be close to regular (but not all the way regular). They first show that direct distribution testing is insufficient: there are irregular distributions where the optimal auction cannot be learned for any fixed number of samples. They put forward a tester, that results in theoretical guarantees that are relative to the distance between the empirical distribution and regular distributions, so as the distance to regularity increases, the guarantees weaken.

They use first-order stochastic dominance to relate the near-regular distribution to all close-by regular distributions. Using this, they show that their approach can be used in three key results: optimal auctions, Bulow and Klemperer's approximation result (add a bidder is better than adding a reserve), and to monopoly pricing.

**Claims And Evidence:**

Yes. The claims of the paper are supported by theoretical proofs and lower bounds.

**Essential References Not Discussed:**

This work mentions Roughgarden and Schrijvers 2016, but should add a little more discussion and differentiation since that approach does give guarantees for irregular distributions.

**Experimental Designs Or Analyses:**

N/A

**Methods And Evaluation Criteria:**

N/A

**Other Comments Or Suggestions:**

n/a

**Other Strengths And Weaknesses:**

In general, the paper is very well written and the three results gives it a breadth of applicability and illustrates well that this technique is broadly applicable within bayesian mechanism design.

The only missing piece (which needs to be there) is the use and discussion of the correct benchmark for irregular settings. With this, it is an accept or strong accept.

When the auction is irregular, the optimal auction is not an optimal reserve by itself, but it includes ironing of the virtual values and allocation rule: the lower value bidders are given a preferred allocation to add competition to the higher valued bidders (Myerson '81). Now, I am almost 100% certain that the revenue from that in a situation where it is almost regular should also be near enough that the theoretical results here should still work against the optimal auctions for irregular distributions that are \alpha close to the empirical distribution since the ironing will be very small if nearly regular. Please add to results if my intuition is correct, or discuss why nearby irregular auctions cannot be included in the benchmark.

**Questions For Authors:**

Do the bounds for the regularized learners also apply to irregular distributions that are \alpha close to D, if they are also \alpha close to a regular distribution?

**Relation To Broader Scientific Literature:**

This work builds on prior work in learning approximately optimal auctions and testing distributions. It makes a contribution by showing that testing for regularity and then operating on that is insufficient, and new results need to be leveraged.

**Theoretical Claims:**

I reviewed briefly the proof for the Bulow-Klemperer result and Theorem 3 and both appear to be correct.

---

> ### Author Rebuttal · Authors · 2025-04-01
>
> Thank you for reading our paper and appreciating our results. We would first like to address the importance of our work in the context of Roughgarden and Schrijvers (2016) and, more generally, Guo et al. (2019) [2]. Both of these papers provide sample complexity results for learning optimal auctions over bounded, irregular distributions, where the learning benchmark is the irregular distribution’s optimal mechanism. These results do not apply to unbounded, irregular distributions. In contrast, our framework can handle this case; we can provide non-trivial revenue guarantees that hold for unbounded, irregular distributions. We will emphasize this important distinction in the next version of our manuscript. Next, we would like to address your question about whether the bounds for the regularized learners should also apply to nearby irregular distributions. These bounds should not apply to nearby irregular distributions. Even if the ironing region is small, nearby irregular distributions may exhibit radically different optimal revenues.
>
> [1] Tim Roughgarden and Okke Schrijvers. “Ironing in the Dark.” Proceedings of the 2016 ACM Conference on Economics and Computation (pp. 1-18) (2016).
>
> [2] Chenghao Guo, Zhiyi Huang, and Xinzhi Zhang. "Settling the sample complexity of single-parameter revenue maximization." Proceedings of the 51st Annual ACM SIGACT Symposium on Theory of Computing (pp. 662-673) (2019).

---

### Decision · Program_Chairs · 2025-05-01

**Decision:**

Accept (poster)

**Comment:**

This paper explores a potentially very interesting and exciting direction: The application of Rubinfeld and Vasilyan's (STOC 2023) "testable learning" framework to auction design. Importantly, they focus on the case of *unbounded* distributions. Unboundedness is important, because there are efficient learning algorithms for bounded irregular distributions (e.g. Roughgarden and Schrijvers 2016, and Guo et al. 2019). Their instantiation for the "testable learning" framework has two parts: (a) a tester which can decide whether a distribution is close to a regular distribution, and (b) in the case where the test is positive, the algorithm finds a mechanism that is close to the best mechanism over all close regular distributions. One thing that their framework does not exclude (and they show this is unavoidable) are irregular distributions which are close to regular distributions (so these distributions pass (a)), but where the revenue on the actual distribution is far more than the optimal revenue on any close-by regular distribution.

Overall we had a good mix of expertise on this paper (from reviewers well-versed in auction theory, and reviewers familiar with the "testable learning" framework). Ultimately we agreed that applying the "testable learning" framework to auction design is a nice idea, but the issue with "false positives" remains (and weakens the result a bit). Roughly the guarantee is: If you cannot be sure that it is not regular, then you may live with the fact that the revenue you get could be suboptimal.